# Cross-Modal Priming Effect of Rhythm on Visual Word Recognition and Its Relationships to Music Aptitude and Reading Achievement

**DOI:** 10.3390/brainsci8120210

**Published:** 2018-11-29

**Authors:** Tess S. Fotidzis, Heechun Moon, Jessica R. Steele, Cyrille L. Magne

**Affiliations:** 1Literacy Ph.D. Program, Middle Tennessee State University, Murfreesboro, TN 37132, USA; 2Institutional Research, University of South Alabama, Mobile, AL 36688, USA; hmoon@southalabama.edu; 3Psychology Department, Middle Tennessee State University, Murfreesboro, TN 37132, USA; jrs2bw@mtmail.mtsu.edu (J.R.S.); cyrille.magne@mtsu.edu (C.L.M.)

**Keywords:** implicit prosody, rhythm sensitivity, event related potentials, reading achievement, musical aptitude

## Abstract

Recent evidence suggests the existence of shared neural resources for rhythm processing in language and music. Such overlaps could be the basis of the facilitating effect of regular musical rhythm on spoken word processing previously reported for typical children and adults, as well as adults with Parkinson’s disease and children with developmental language disorders. The present study builds upon these previous findings by examining whether non-linguistic rhythmic priming also influences visual word processing, and the extent to which such cross-modal priming effect of rhythm is related to individual differences in musical aptitude and reading skills. An electroencephalogram (EEG) was recorded while participants listened to a rhythmic tone prime, followed by a visual target word with a stress pattern that either matched or mismatched the rhythmic structure of the auditory prime. Participants were also administered standardized assessments of musical aptitude and reading achievement. Event-related potentials (ERPs) elicited by target words with a mismatching stress pattern showed an increased fronto-central negativity. Additionally, the size of the negative effect correlated with individual differences in musical rhythm aptitude and reading comprehension skills. Results support the existence of shared neurocognitive resources for linguistic and musical rhythm processing, and have important implications for the use of rhythm-based activities for reading interventions.

## 1. Introduction

Music and language are complex cognitive abilities that are universal across human cultures. Both involve the combination of small sound units (e.g., phonemes for speech, and notes for music) which in turn, allow us to generate an unlimited number of utterances or melodies, in accordance with specific linguistic or musical grammatical rules (e.g., [1]). Of specific interest for the present study, is the notion of rhythm. In music, rhythm is marked by the periodic succession of acoustic elements as they unfold over time, and some of these elements may be perceived as stronger than others. Meter is defined as the abstract hierarchical organization of these recurring strong and weak elements that emerge from rhythm. It is this metrical structure that allows listeners to form predictions and anticipations, and in turn dance or clap their hands to the beat of the music [2]. 

Similarly, in speech, the pattern of stressed (i.e., strong), and unstressed (i.e., weak) syllables occurring at the lexical level contributes to the metrical structure of an utterance. Lexical stress is usually defined as the relative emphasis that one syllable, or several syllables, receive in a word [3]. Stress is typically realized by a combination of increased duration, loudness, and/or pitch change. In many languages, such as English, the salience of the stressed syllable is further reinforced by the fact that many unstressed syllables contain a reduced vowel [4]. Some languages are described as having fixed stress because the location of the stress is predictable. For instance, in French, the stress is usually on the final full syllable [5]. By contrast, several languages are considered to have variable stress because the position of the stress is not predictable. In such languages, like English, stress may serve as a distinctive feature to distinguish noun-verb stress homographs [6]. For example, the word “permit” is stressed on the first syllable when used as a noun, but stressed on the second syllable when used as a verb.

There is increasing support for the existence of rhythmic regularities in English, despite the apparent lack of physical periodicity of the stressed syllables when compared to the rhythmic structure of music (e.g., [7]). During speech production, rhythmic adjustments, such as stress shifts, may take place to avoid stress on adjacent syllables, and these stress shifts may give rise to a more regular alternating pattern of stressed and unstressed syllables [8]. For example, “thirteen” is normally stressed on the second syllable, but the stress can shift to the first syllable when followed by a word with initial stress (e.g., “thirteen people”). These rhythmic adjustments may play a role in speech perception, as suggested by findings showing that sentences with stress shifts are perceived as more natural than sentences with stress clashes, despite that words with shifted stress deviate from their default metrical structure [9]. 

In music, the Dynamic Attending Theory (DAT) provides a framework in which auditory rhythms are thought to create hierarchical expectancies for the signal as it unfolds over time [10,11]. According to the DAT, distinct neural oscillations entrain to the multiple hierarchical levels of the metrical structure of the auditory signal, and strong metrical positions act as attentional attractors, thus making acoustic events occurring at these strong positions easier to process. Similarly, listeners do not pay equal attention to all parts of the speech stream, and speech rhythm may influence which moments are hierarchically attended to in the speech signal. For instance, detection of a target phoneme was found to be faster if it was embedded in a rhythmically regular sequence of words (i.e., regular time interval between successive stressed syllables), thus suggesting that speech rhythm cues, such as stressed syllables, guide listeners’ attention to specific portions of the speech signal [12]. Further evidence suggests that predictions regarding speech rhythm and meter may be crucial for language acquisition [13], speech segmentation [14], word recognition [15], and syntactic parsing [16].

Given the structural similarities between music and language, a large body of literature has documented which neuro-cognitive systems may be shared between language and music (e.g., [7,17,18]), and converging evidence support the idea that musical and linguistic rhythm perception skills partially overlap [19,20,21]. In line with these findings, several EEG studies revealed a priming effect of musical rhythm on spoken language processing. For instance, listeners showed a more robust neural marker of beat tracking and better comprehension when stressed syllables aligned with strong musical beats in sung sentences [22]. Likewise, EEG findings demonstrated that spoken words were more easily processed when they followed non-linguistic primes with a metrical structure that matched the word metrical structure [23]. A follow-up study using a similar design showed this benefit of rhythm priming on speech processing may be mediated by cross-domain neural phase entrainment [24].

The purpose of the present study was to shed further light on the effect of non-linguistic rhythmic priming on language processing (e.g., [22,23,24]). We specifically focused on words with a trochaic stress pattern (i.e., a stressed syllable followed by an unstressed syllable) because in the English lexicon, they constitute more than 85% of content words [25]. This high frequency of the trochaic pattern may play a particularly preponderant role in English language development, as infants seem to adopt a metrical segmentation strategy by considering a stress syllable as the beginning of a word in the continuous speech stream [26]. Evidence in support of this important role of the trochaic pattern comes from studies conducted with English speaking infants who develop a preference for the trochaic pattern as early as the age of 6 months [27]. By contrast, the ability to detect words with an iambic pattern (i.e., an unstressed syllable followed by a stressed syllable) develops later, around 10.5 months, and seems to rely more on using additional sets of linguistic knowledge regarding phonotactic constraints (i.e., the sequences of phonemes that are allowed in a given language), and allophonic cues (i.e., the multiple phonetic variants of a phoneme, whose occurrences depend on their position in a word and their phonetic context), rather than stress cues [13].

The first specific aim was to examine whether the cross-domain rhythmic priming effect is also present when target words are visually presented. To this end, participants were presented with rhythmic auditory prime sequences (either a repeating pattern of long-short or short-long tone pairs), followed by a visual target word with a stress pattern that either matched, or mismatched, the temporal structure of the prime (See Figure 1). Based on previous literature (e.g., [20,23,28]), we predicted that words that do not match the temporal structure of the rhythmic prime would elicit an increased centro-frontal negativity. 

A second aim of the study was to determine whether such rhythmic priming effect would be related to musical aptitude. Musical aptitude has been associated with enhanced perception of speech cues that are important correlates of rhythm. For instance, individuals with formal musical training better detect violations of word pitch contours [29,30] and syllabic durations [31] than non-musicians. In addition, electrophysiological evidence shows that the size of a negative ERP component elicited by spoken words with an unexpected stress pattern correlates with individual differences in musical rhythm abilities [20]. Thus, in the present study, we expected the amplitude of the negativity elicited by the cross-modal priming effect to correlate with individual scores on a musical aptitude test, if the relationship between musical aptitude and speech rhythm sensitivity transfers to the visual domain. 

Finally, the third study aim was to test whether the cross-modal priming effect present in the ERPs correlated with individual differences in reading achievement. Mounting evidence suggests a link between sensitivity to auditory rhythm skills (both linguistic and musical) and reading abilities (e.g., [32,33,34,35]). As such, we collected individuals’ scores on a college readiness reading achievement test to examine whether the cross-modal ERP effect correlated with individual differences in reading comprehension skills. We expected the amplitude of the negativity elicited by the cross-modal priming effect to correlate with individual scores on the American College Testing (ACT) reading test, if rhythm perception skills relate to reading abilities as suggested by the current literature [32,33,34,35].

## 2. Materials and Methods

### 2.1. Participants

Eighteen first year college students took part in the experiment (14 females and 4 males, mean age = 19.5, age range: 18–22). All were right-handed, native English speakers with less than two years of formal musical training. None of the participants were enrolled in a Music major. The study was approved by the Institutional Review Board at Middle Tennessee State University, and written consent was obtained from the participants prior to the start of the experiment. 

### 2.2. Standardized Measures

The Advanced Measures of Music Audiation (AMMA; [36]) was used to assess participants’ musical aptitude. The AMMA has been used previously to measure the correlation between musical aptitude and index of brain activities (e.g., [20,37,38,39]). This measure was nationally standardized with a normed sample of 5336 U.S. students and offers percentile ranked norms for both music and non-music majors. Participants were presented with 30 pairs of melodies and asked to determine whether the two melodies of each pair were the same, tonally different, or rhythmically different. The AMMA provides separate scores for rhythmic and tonal abilities. For non-Music majors, reliability scores are 0.80 for the tonal score and 0.81 for the rhythm score [36].

The reading scores on the ACT exam were used to examine the relationship between reading comprehension and speech rhythm sensitivity. The ACT reading section is a standardized achievement test that comprises short passages from four categories (prose fiction, social science, humanities, and natural science) and 40 multiple-choice questions that test the reader’s comprehension of the passages. Scores range between 1 and 36. The test was administered and scored by the non-profit organization of the same name (ACT, Inc., Iowa City, IA, USA) using a paper and pencil format.

### 2.3. EEG Cross-Modal Priming Paradigm

Prime sequences consisted of a rhythmic tone pattern of either a long-short or short-long structure repeated three times. The tones consisted of a 500 Hz sine wave with a 10 ms rise/fall, and a duration of either 200 ms (long) or 100 ms (short). In long-short sequences, the long tone and short tone were separated by a silence of 100 ms, and each of the three successive long-short tone pairs was followed by a silence of 200 ms. In short-long sequences, the short tone and long tone were separated by a silence of 50 ms, and each of the three successive short-long tone pairs was followed by a silence of 250 ms. Because previous research has shown that native speakers of English have a cultural bias toward grouping a sequence of tones differing in duration, into short-long patterns [40,41], a series of behavioral pilot experiments were conducted with different iterations of the tone sequences to determine which parameters would provide consistent perception of either long-short or short-long patterns.

Visual targets were composed of 140 English real-word bisyllabic nouns and 140 pseudowords, which were all selected from the database of the English Lexicon Project [42]. The lexical frequency of all the words was controlled using the log HAL frequency [43]. The mean log HAL frequency for each set of stress patterns was 10.28 (SD = 0.98) for trochaic sequences and 10.28 (SD = 0.97) for iambic sequences. Pseudowords were matched to the real words in terms of syllable count and word length and were used only for the purpose of the lexical decision task. Half of the real words (*N* = 70) had a trochaic stress pattern (i.e., stressed on the first syllable, for example, “basket”). The other half consisted of fillers with an iambic stress pattern (i.e., stressed on the second syllable, for example, “guitar”). 

Short-long and long-short prime sequences were combined with the visual target words to create two experimental conditions in which the stress pattern of the target word either matched or mismatched the rhythm of the auditory prime.

We chose to analyze only the ERPs elicited by trochaic words for several reasons. First, trochaic words comprise the predominant stress pattern in English (85–90% of spoken English words according to [34]), and consequently, participants will likely be familiar with their pronunciation. Second, because stressed syllables correspond to word onset in trochaic words, this introduces fewer temporal jitters than for iambic words when computing ERPs across trials. This scenario is particularly problematic for iambic words during silent reading, because there is no direct way to measure when participants read the second syllable. Third, participants were recruited from a university located in the southeastern region of the United States, and either originated from this area, or have been living in the area for several years. It is well documented that the Southern American English dialect tends to place stress on the first syllable of many iambic words despite that these types of words are stressed on the second syllable in standard American English (e.g., [44]). As such, rhythmic expectations are less clear to predict for iambic words.

### 2.4. Procedure

Participants’ musical aptitude was first measured using the AMMA [36]. Following administration of the AMMA test, participants were seated in a soundproofed and electrically shielded room. Auditory prime sequences were presented through headphones, and target stimuli were visually presented on a computer screen placed at approximately 3 feet in front of the participant. Words and pseudowords were written in black lowercase characters on a white background. No visual cue was provided to the participant regarding the location of the stress syllables in the target words. Stimulus presentation was controlled using the software E-prime 2.0 Professional with Network Timing Protocol (Psychology software tools, Inc., Pittsburgh, PA, USA). Participants were presented with 5 blocks of 56 stimuli. The trials were randomized within each block, and the order of the blocks was counterbalanced across participants. Each trial was introduced by a fixation cross displayed at the center of a computer screen that remained until 2 s after the onset of the visual target word. Participants were asked to silently read the target word and to press one button if they thought it was a real English word, or another button if they thought it was a nonword. The entire experimental session lasted 1.5 h.

### 2.5. EEG Acquisition and Preprocessing

EEG was recorded continuously from 128 Ag/AgCL electrodes embedded in sponges in a Hydrocel Geodesic Sensor Net (EGI, Eugene, OR, USA) placed on the scalp, connected to a NetAmps 300 amplifier, and using a MacPro computer. Electrode impedances were kept below 50 kΩ. Data was referenced online to Cz and re-referenced offline to the averaged mastoids. In order to detect the blinks and vertical eye movements, the vertical and horizontal electrooculograms (EOG) were also recorded. The EEG and EOG were digitized at a sampling rate of 500 Hz. EEG preprocessing was carried out with NetStation Viewer and Waveform tools. The EEG was first filtered with a bandpass of 0.1 to 30 Hz. Data time-locked to the onset of trochaic target words was then segmented into epochs of 1100 ms, starting with a 100 ms prior to the word onset and continuing 1000 ms post-word-onset. Trials containing movements, ocular artifacts, or amplifier saturation were discarded. ERPs were computed separately for each participant and each condition by averaging together the artifact-free EEG segments relative to a 100 ms pre-baseline. 

### 2.6. Data Analysis

Statistical analyses were performed using MATLAB and the FieldTrip open source toolbox [45]. A planned comparison between the ERPs elicited by mismatching trochaic words and matching trochaic words was performed using a cluster-based permutation approach. This non-parametric data-driven approach does not require the specification of any latency range or region of interest a priori, while also offering a solution to the problem of multiple comparisons (see [46]).

To relate the ERP results to the behavioral measures (i.e., musical aptitude and reading comprehension), an index of sensitivity to speech rhythm cues was first calculated from the ERPs using the mean of the significant amplitude differences between ERPs elicited by matching and mismatching trochaic words at each channels, and time points belonging to the resulting clusters (see [20,47] for similar approaches). Pearson correlations were then tested between the ERP cluster mean difference and the participants’ scores on the AMMA and ACT reading section, respectively. A multiple regression was also computed with the ERP cluster mean difference as the outcome measure, and the AMMA Rhythm scores and ACT Reading scores as the predictor variables.

## 3. Results

### 3.1. Metrical Expectancy

Overall, participants performed well on the lexical decision task, as suggested by the mean accuracy rate (*M* = 98.82%, SD = 0.85). A paired samples *t*-test was computed to compare accuracy rates for real target words in the matching (*M* = 99.83%, SD = 0.70), and mismatching (*M* = 99.42%, SD = 1.40) rhythm conditions. No statistically significant differences were found between the two conditions, *t* (35) = 1.54, *p* = 0.13, two-tailed.

Analyses of the ERP data revealed that target trochaic words that mismatched the rhythmic prime elicited a significantly larger negativity from 300 to 708 ms over a centro-frontal cluster of electrodes (*p* < 0.001, See Figure 2).

### 3.2. Brain-Behavior Relationships

The negative ERP cluster mean difference was statistically significantly positively correlated with the AMMA Rhythm scores (*r* = 0.74, *p* < 0.001; see Figure 3A) and the ACT Reading scores (*r* = 0.60, *p* = 0.009; see Figure 3B). A statistically significant positive correlation was also found between the AMMA Rhythm scores and ACT Reading scores (*r* = 0.55, *p* = 0.016; see Figure 3C). By contrast, no statistically significant correlation was found between the AMMA Tonal scores and the negative ERP cluster mean difference (*r* = 0.30, *p* = 0.23) or the ACT Reading scores (*r* = 0.09, *p* = 0.70). The maximum Cook’s distance for the reported correlations indicated no undue influence of any data point on the fitted models (max Cook’s *d* < 0.5).

A multiple regression was conducted to investigate whether AMMA Rhythm scores and ACT Reading scores predicted the size of the negative ERP cluster mean difference. Table 1 summarizes the analysis results. The regression model explained 59.9% of the variance and was a statistically significant predictor of the negative ERP cluster mean difference (*R*^2^ = 0.599, *F* (2,15) = 11.2, *p* = 0.001). As can be seen in Table 1, AMMA Rhythm scores statistically significantly contributed to the model (*β* = 0.594, *t* (15) = 3.023, *p* = 0.009), but ACT Reading scores did not (*β* = 0.267, *t* (15) = 1.359, *p* = 0.194). The final predictive model was: Negative ERP Cluster Mean Difference = (0.281 × AMMA Rhythm) + (0.081 × ACT Reading) + 8.000.

## 4. Discussion

The current study aimed to examine the cross-modal priming effect of non-linguistic auditory rhythm on written word processing and investigate whether such effect would relate to individual differences in musical aptitude and reading comprehension. As hypothesized, trochaic target words that did not match the rhythmic structure of the auditory prime were associated with an increased negativity over the centro-frontal part of the scalp. This finding is in line with previous ERP studies on speech rhythm and meter [6,15,20,28,31,48,49,50]. It has been generally proposed that this negative effect either reflects an increased N400 [15,49], or a domain-general rule-based error-detection mechanism [6,20,28,31,51,52]. The fact that similar negative effects have been reported in response to metric deviations in tone sequences (e.g., [53,54]) further supports the latter interpretation. 

While the aforementioned studies were conducted either in the linguistic or musical domain, the negative effect observed for mismatching target words was generated by non-linguistic prime sequences in the present experiment. Cason and Schön [23] previously reported a cross-domain priming effect of music on speech processing, which was reflected by a similar increased negativity when the metrical structure of the spoken target word did not match the rhythmic structure of the musical prime. Several other findings have since shown that temporal expectancies generated by rhythmically regular non-linguistic primes can facilitate spoken language processing in typical adults (e.g., [24,55]), and children [56,57], as well as adults with Parkinson’s disease [58], children with cochlear implants [59], and children with language disorders [60]. This beneficial effect may stem from the regular rhythmic structure of the prime, which provides temporally predictable cues to which internal neural oscillators can anchor [24]. The present findings support and extend this line of research by showing this negativity is elicited even when the target words were visually presented, thus suggesting that non-linguistic rhythm can not only induce metrical expectations across distinct cognitive domains, but also across different sensory modalities [61]. These findings also provide additional evidence in favor of the view that rhythm/meter processing relies on a domain-general neural system that is not specific to language [19,21,22].

We further investigated whether this cross-modal priming effect was related to individual differences in musical aptitude. Interestingly, our results showed a statistically significant correlation between the size of the brain response elicited by unexpected stress patterns and the AMMA rhythm subscore, but not the tonal subscore. In addition, musical rhythm aptitude was a statistically significant predictor of speech rhythm sensitivity, even after controlling for reading comprehension skills. This is in line with previous ERP studies showing that adult musicians performed better than non-musicians at detecting words pronounced with an incorrect stress pattern [31]. In addition, this enhanced sensitivity to speech meter was associated with larger electrophysiological responses to incorrectly pronounced words, which was interpreted as reflecting more efficient early auditory processing of the temporal properties of speech. 

Robust associations have also been found between musical rhythm skills and speech prosody perception, even after controlling for years of music education [19]. Noteworthy for the present experiment, individual differences in brain sensitivity to speech rhythm variations can be explained by variance in musical rhythm aptitude in individuals with less than two years of musical training. For instance, in a recent experiment [20], participants’ musical aptitude was assessed using the same standardized measure of musical abilities (i.e., AMMA) as in the present study. Participants listened to sequences consisting of four bisyllabic words for which the stress pattern of the final word either matched or mismatched the stress pattern of the preceding words. Words with a mismatching stress pattern elicited an increased negative ERP component with the same scalp distribution and latency as the one found in the current data. More importantly, participants’ musical rhythm aptitude statistically significantly correlated with the size of the negative effect. Thus, in light of the aforementioned literature, the present results confirm and extend previous data suggesting a possible transfer of learning between the musical and linguistic domains (See [62] for a review).

Adding to the growing literature showing a relationship between sensitivity to speech rhythm and reading skills, our results revealed a statistically significant positive correlation between the scores on the ACT reading subtest and the size of the negative ERP effect elicited by mismatching stress patterns. Previous studies have mainly focused on typically developing young readers using several novel speech rhythm tasks in conjunction with standardized measures of reading abilities, and results consistently showed a correlation between performances on the speech rhythm tasks and individual differences in word reading skills [63,64,65,66]. It has been proposed that early sensitivity to speech rhythm cues may contribute to the development of phonological representations [32]. However, sensitivity to speech rhythm cues still explains unique variance in word reading skills after controlling for phonological processing skills [67], thus suggesting that it also makes a significant contribution to reading development independently of phonological awareness.

More directly related to the present study, research with older readers and adults suggests that knowledge of the prosodic structure of words continues to play a role in skilled reading. For instance, visual word recognition is facilitated when primed by word fragments with a matching stress pattern [68,69]. Two other studies conducted on typical adults focused on lexical stress perception in isolated multisyllabic words [70,71], and found a significant relationship with reading comprehension. Likewise, adult struggling readers usually show lower performance than their typical peers on tasks measuring perception of word stress patterns or auditory rhythms [72,73,74,75] (but see [74,76]).

Interestingly, the finding that reading comprehension was not a statistically significant contributor to speech rhythm sensitivity after controlling for musical rhythm aptitude supports the Temporal Sampling Framework (TSF) proposed by Goswami [32]. According to the TSF, the link between speech rhythm sensitivity and reading skills is mediated by domain-general neurocognitive mechanisms for processing acoustic information carrying rhythmic cues. In line with this interpretation, we found a statistically significant correlation between the AMMA rhythm scores and reading achievement scores. 

The OPERA (overlap, precision, emotion, repetition, attention) hypothesis formulated by Patel [77,78] further provides a potential explanation of music-training driven plasticity in brain networks involved in language. OPERA offers a set of five optimal conditions that must be met for music training to drive plasticity: (1) music and language have overlapping anatomical substrates; (2) music activities require a greater level of precision compared to language; (3) music activities evoke strong emotions; (4) music training involves repeated practice; (5) music activities require sustained attention. In line with this framework, the Precise Auditory Timing Hypothesis (PATH) proposed by Tierney and Kraus [79] predicts that music programs that focus on rhythm activities, with an emphasis on entrainment and timing, will be more effective in improving reading-related skills, such as phonological processing skills, because there are overlaps between language and music networks processing rhythmic information, and music requires a higher level of auditory-motor timing precision than language. OPERA and PATH thus provide compelling explanations for the significant relationships we report here between musical rhythm aptitude, speech rhythm sensitivity, and reading achievement. While our present study was correlational (and conducted with non-musicians), data from recent longitudinal studies using randomized controlled trials indeed show promising results of rhythm-based intervention for the development of language skills in children with reading disorders [80], and typical peers [81].

Finally, the fact that we found a “metrical” negativity to visual targets, despite that participants were not allowed to sound out the words, further supports theories proposing that information about the metrical structure of a word is part of its lexical representation and automatically retrieved during silent reading [82,83]. This idea is in line with the Implicit Prosody Hypothesis (IPH) originally proposed by Fodor [84]. The IPH is closely related to the concept of verbal imagery or inner voice, which can be found in the literature throughout the 20th century [82]. According to the IPH, readers create a mental representation of the prosodic structure of the text while they are silently reading. Several studies have provided compelling evidence in support for the IPH, especially regarding lexical stress. For instance, eye-tracking studies showed that readers had longer reading times and more eye fixations for four-syllable words with two stressed syllables, than for one stressed syllable [85], and that expectations generated by the stress pattern of successive words may affect early stages of syntactic analysis of upcoming words in written sentences [82,86]. Taken together, these results and the present data provide compelling evidence for a role of prosodic representations regarding a word stress pattern during silent reading. 

One potential limitation of the current research is the use of ACT reading scores, which may not be fully representative of the participants’ reading skills. In particular, phonemic awareness, decoding, and fluency, which are components known to greatly contribute to reading comprehension [87], cannot be teased apart in the ACT reading subsets. Future research using a more comprehensive battery of language and reading assessments would better allow a more complete understanding of which reading components are more closely related to speech rhythm perception skills.

## 5. Conclusions

The present data confirm and extend previous studies showing facilitating effects of a regular non-linguistic rhythm on spoken language processing (e.g., [23,55,59]), by demonstrating this to also be the case for written language processing. We propose that this cross-modal effect of rhythm is mediated by the automatic retrieval of the word metrical structure (i.e., implicit prosody) during silent reading (i.e., implicit prosody generated through verbal imagery). Finally, because we found that the negativity associated with this cross-modal priming effect of rhythm correlated with individual differences in musical aptitude and reading achievement, this further supports the potential clinical and education implications of using rhythm-based intervention for populations with language or learning disabilities.

## Figures and Tables

**Figure 1 brainsci-08-00210-f001:**
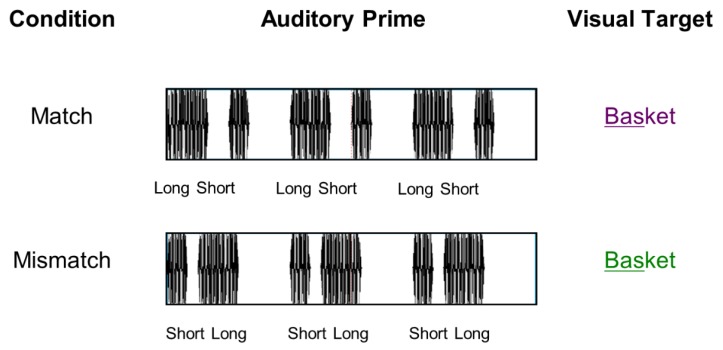
Rhythmic cross-modal priming experimental paradigm. The auditory prime (long-short or short-long sequence) is followed by a target visual word with a stress pattern that either match or mismatch the prime (Note: stressed syllable is underlined for illustration purposes only).

**Figure 2 brainsci-08-00210-f002:**
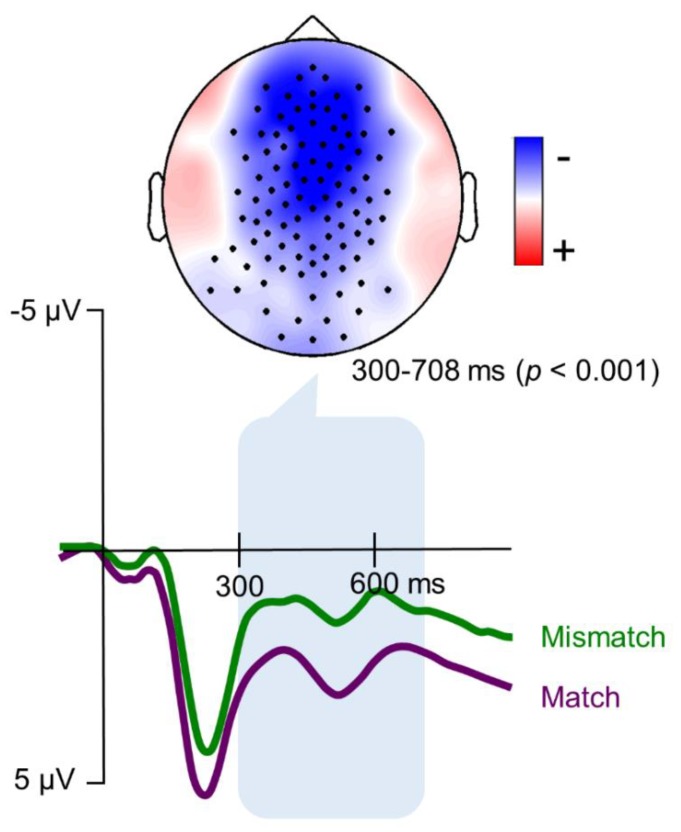
Rhythmic priming Event-related potential (ERP) effect. Grand-average event-related potentials (ERPs) recorded for matching (purple), and mismatching (green) trochaic target words, averaged for the significant group of channels in the cluster. The latency range of the significant clusters is indicated in blue. (Note: Negative amplitude values are plotted upward. The topographic map shows the mean differences in scalp amplitudes in the latency range of the significant clusters. Electrodes belonging to the cluster are indicated with a black dot).

**Figure 3 brainsci-08-00210-f003:**
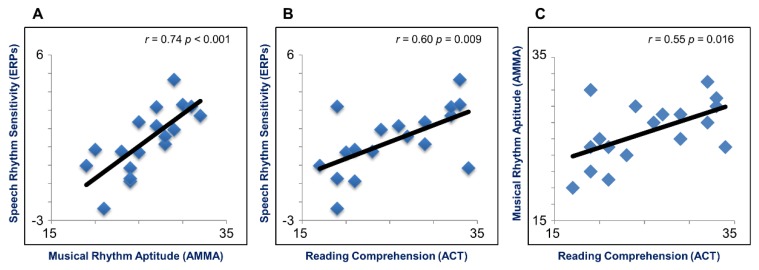
Brain-behavior correlations. (**A**) Correlation between speech rhythm sensitivity (as indexed by the negative ERP cluster mean difference) and musical rhythm aptitude; (**B**) correlation between speech rhythm sensitivity and reading comprehension; (**C**) correlation between musical rhythm aptitude and reading comprehension. (Note: The solid line represents a linear fit.)

**Table 1 brainsci-08-00210-t001:** Multiple regression coefficients.^1^

Source	*B*	*SE*	*β*	*t*	*p*
Constant	8.000	2.033		3.935	0.001
ACT Reading	0.081	0.060	0.267	1.359	0.194
AMMA Rhythm	0.281	0.093	0.594	3.023	0.009

^1^ Outcome: Negative ERP cluster mean difference; *B*: unstandardized coefficient; *SE*: standard error; *β*: standardized coefficient; *t*: *t*-value; *p*: *p*-value; ACT: American College Testing; AMMA: Advanced Measures of Music Audiation.

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
