# Peer review of "Cross-Modal Priming Effect of Rhythm on Visual Word Recognition and Its Relationships to Music Aptitude and Reading Achievement"

_brainsci, 2018, doi:10.3390/brainsci8120210_

Reviewer 1 Report

Thanks for the opportunity to review this submission.

I believe the work represents and important incremental finding regarding the role of metrical priming on lexical processing, and of the relationship between music and language.

I have only three primary concerns, which I believe could be addressed through additional discussion by the authors, along with several minor concerns, listed below:

1) It is not entirely clear that the priming stimuli used in this experiment constitute *music*, being simple two-tone sequences differing in length, with constant pitch. It could be that these stimuli require only a basic (nonmusical, nonlinguistic) temporal/rhythmic processing and that this basic rhythmic processing is correlated both with the linguistic task and the AMMA measures. Some discussion of this would clarify (see, eg, Patel 2011)

2) I have some serious questions about the word stimuli, as well. I understand the imbalance between the trichaic and iambic examples, based on the English lexicon, but more information is needed.

a) were the iambic stimuli analyzed at all, and if so, what did they show? (you could at least attempt to analyze them using a fixed average syllable length)

b) were iambs and trochees comparable in terms of lexical frequency?

c) did you filter out noun-verb pairs which could be either iambic or trochaic? (e.g., 'record')

3) the lack of an omnibus statistical test in section 3.2 is pretty glaring - we do not know to what degree the ACT and AMMA scores were correlated; thus, we are not able to consider the totality of effects on the ERP components; these could be combined in a regression analysis

minor concerns

ln 52 - needs clarification that this is a description only of ENglish - some discussion of crosslinguistic differences in stress systems (e.g., fixed, free) would be appropriate here.

100 the hypothesis is not stated for the 3rd aim.

104 the term 'freshman' is out of vogue, at least at my university - consider 'first year'

104 # of females are reported, but this does not entail that the remainder of participants were male

116 how was ACT information obtained, and do you know how it was administered? (paper, electronic)

Author Response

November 14, 2018

Dear Editors,

Please find attached our revisions for the manuscript entitled “Cross-Modal Priming Effect of Rhythm on Visual Word Recognition and its Relationships to Music Aptitude and Reading Achievement”.  We are pleased that Reviewer 1 found that our work represented an  important incremental finding regarding the role of metrical priming on  lexical processing as well as the relationship between music and  language.

Please  find below our detailed responses to the reviewer’s comments. We hope  to have addressed all of your concerns and that you will find the  revised manuscript suitable for publication.

Sincerely yours,

Tess Fotidzis, Heechun Moon, Jessica Steele, Cyrille Magne

Reviewer #1

Thanks for the opportunity to review this submission.

I  believe the work represents an important incremental finding regarding  the role of metrical priming on lexical processing, and of the  relationship between music and language.

I  have only three primary concerns, which I believe could be addressed  through additional discussion by the authors, along with several minor  concerns, listed below:

1.     It  is not entirely clear that the priming stimuli used in this experiment  constitute *music*, being simple two-tone sequences differing in length,  with constant pitch. It could be that these stimuli require only a  basic (nonmusical, nonlinguistic) temporal/rhythmic processing and that  this basic rhythmic processing is correlated both with the linguistic  task and the AMMA measures. Some discussion of this would clarify (see,  eg, Patel 2011)

Response:  We agree with the reviewer that the use of the term “music” in regard  to the prime stimuli may be too strong in the present study. We now use  the term “non-linguistic rhythm sequences” throughout the manuscript. We  have also expanded the discussion of our present results in light of  the OPERA hypothesis proposed by Patel (2011; 2014); see page 8, lines  306-320.

2.     I  have some serious questions about the word stimuli, as well. I  understand the imbalance between the trochaic and iambic examples, based  on the English lexicon, but more information is needed.

a)  were the iambic stimuli analyzed at all, and if so, what did they show?  (you could at least attempt to analyze them using a fixed average  syllable length)

b) were iambs and trochees comparable in terms of lexical frequency?

c) did you filter out noun-verb pairs which could be either iambic or trochaic? (e.g., 'record')

Response: We are thankful to the Reviewer for these insightful remarks that helped make the Introduction and Methods clearer.

a)     We  did not analyze the ERP responses to iambic words since they were  intended to be used as fillers, while the purpose of the study and  associated hypotheses were focused specifically on the trochaic pattern.  We specifically focused on words with a trochaic stress pattern (i.e., a  stressed syllable followed by an unstressed syllable) because in the  English lexicon, they constitute more than 85% of content words (Cutler  & Carter, 1987).  This high frequency of the trochaic pattern may  play a particularly preponderant role in English language development,  as infants seem to adopt a metrical segmentation strategy by considering  a stress syllable as the beginning of a word in the continuous speech  stream (Cutler & Norris, 1988).  Evidence in support of this  important role of the trochaic pattern comes from studies conducted with  English speaking infants who develop a preference for the trochaic  pattern as early as the age of 6 months (Jusczyk et al., 1993).  By  contrast, the ability to detect words with an iambic pattern (an  unstressed syllable followed by a stressed syllable) develops later,  around 10.5 months, and seems to rely more on the need for developing  and using additional sets of linguistic knowledge regarding phonotactic  constraints (i.e., the sequences of phonemes that are allowed in a given  language), and allophonic cues (i.e., the multiple phonetic variants of  a phoneme, whose occurrences depend on their position in a word and  their phonetic context), rather than stress cues (Jusczyk, 1999). We  have clarified the purpose of the present experiment by including this  information in the introduction on pages 2-3, lines 88-101.

We  also believe that, from a methodological perspective, determining an  average syllable length is not possible given that target words were  visually presented and reading speeds vary extensively in adult readers.  Computing the averaged syllable length would have necessitated  participants to read the words aloud and to record their oral  production. However, we specifically required our participants to not  sound out the words and read them silently, because we were interested  in the automatic retrieval of prosodic information during silent reading  (see page 9, lines 348-359 in the discussion section).

b)     Since  the task was a lexical decision, we carefully controlled that all real  words were of high lexical frequency in English and that the mean  lexical frequency was equivalent for iambic and trochaic words. The mean  log HAL frequency was 10.28 for trochaic words (SD = 0.98), and also 10.28 for iambic words (SD = 0.97). This information has been added to the Methods section on page 4, lines 165-168.

c)     Regarding  avoiding the use of noun-verb stress homographs, this is a very  important point raised by the Reviewer that we carefully controlled for.  None of the target words used in the present experiment were noun-verb  stress homographs. In fact, one of the previous studies published by two  of the co-authors (Moon & Magne, 2015) showed that noun-verb stress  homographs pronounced with a stress pattern that did not match their  grammatical function in spoken sentences elicited similar negativities  as the ones reported in the present study. In the present study, we thus  excluded them to avoid this potential confounding variable.

3.     the  lack of an omnibus statistical test in section 3.2 is pretty glaring -  we do not know to what degree the ACT and AMMA scores were correlated;  thus, we are not able to consider the totality of effects on the ERP  components; these could be combined in a regression analysis

Response:  We did not initially include the correlation between AMMA scores and  ACT scores because it was not directly testing any of our three  hypotheses. After careful review of the manuscript, we understand this  missing correlation may be confusing for the readers and we have decided  to report it in the Results section (page 7, lines 252-256). Figure 3C  now includes the scatter plot and trendline for this correlation as  well. We also interpret the results of this correlation in the  discussion on page 8, lines 306-320. Please note that we chose to report  separate correlations rather than one single multiple regression  because the outcome measures are different for hypotheses 2 and 3  (speech rhythm sensitivity and reading achievement, respectively).

Minor Concerns:

4.     ln  52 - needs clarification that this is a description only of ENglish -  some discussion of crosslinguistic differences in stress systems (e.g.,  fixed, free) would be appropriate here.

Response:  In line with the Reviewer’s insightful suggestion, we have modified  this section of the Introduction to clarify that it is for English (page  2, line 53), and to discuss the stress system in languages with fixed  stress versus variable stress (pages 1-2, lines 42-52).

5.     100 the hypothesis is not stated for the 3rd aim.

Response:  We thank the Reviewer for catching this inadvertent oversight. The  hypothesis has been added in the introduction on page 3, lines 125-127

6.     104 the term 'freshman' is out of vogue, at least at my university - consider 'first year'

Response: “freshman students” was changed to “first year college students” on page 4, line 130.

7.     104 # of females are reported, but this does not entail that the remainder of participants were male

Response:  The participants specified their gender orally to the experimenter.  Fourteen participants self-identified as female and the remaining four  participants as male. This information has been added on page 4, line  131.

8.     116 how was ACT information obtained, and do you know how it was administered? (paper, electronic)

Response:  The test was administered and scored by the non-profit organization of  the same name (ACT, Inc.) using a paper and pencil format. This  information has been added on page 4, lines 149-151.

Reviewer 2 Report

This paper describes an experiment that tested whether auditory rhythmic primes would influence people's detection of mismatching stress patterns in visually presented words. EEG revealed increased fronto-central negativity in response to mismatched stress patterns, and the magnitude of the negative response correlated with ACT reading scores and scores on the rhythmic component of a musical aptitude test. The study shows that musical rhythmic priming influences visual word processing during silent reading, building on prior research showing similar effects for spoken word processing.

The experiment was well-designed and has been very clearly presented. I have no major concerns, but list a few minor points below:

2.3, 1st paragraph. Were these patterns previously tested for their strength of metricality? It might be worth adding a few sentences justifying their design. In particular, why didn't the structure of the two sequences parallel each other? That is, why was the short-long sequence not 100 ms tone - 100 ms rest - 200 ms tone - 200 ms rest?

2.3, 4th paragraph. Why not include words that are also pronounced with iambic stress in Southern American English and/or run a pronunciation check with the participants? It's not clear why iambic words were included in the set of visual targets at all, if they were not going to be analysed, and showing the same effect for trochaic and iambic words would have strengthened the conclusions.

4, 2nd paragraph. Even if participants were instructed not to read the words out loud, they presumably used verbal imagery to say them covertly/audiate. The way the words were presented (one word at a time, with the stressed syllable underlined) likely encouraged this way of reading. So, despite the visual presentation, the processing of the words was likely still auditory. It seems to me that the experiment shows how musical rhythm priming effects generalize to verbal imagery rather how they generalize to the visual modality. 

Lines 231-232, "...observed for mismatching target *words* was generated by musical prime *sequences*..."

Author Response

November 14, 2018

Dear Editors,

Please find attached our revisions for the manuscript entitled “Cross-Modal Priming Effect of Rhythm on Visual Word Recognition and its Relationships to Music Aptitude and Reading Achievement”. We are pleased that Reviewer 2 found that the experiment was well designed and has been very clearly presented.

Please  find below our detailed responses to the reviewers’ comments. We hope  to have addressed all of your concerns and that you will find the  revised manuscript suitable for publication.

Sincerely yours,

Tess Fotidzis, Heechun Moon, Jessica Steele, Cyrille Magne

Reviewer #2

This  paper describes an experiment that tested whether auditory rhythmic  primes would influence people's detection of mismatching stress patterns  in visually presented words. EEG revealed increased fronto-central  negativity in response to mismatched stress patterns, and the magnitude  of the negative response correlated with ACT reading scores and scores  on the rhythmic component of a musical aptitude test. The study shows  that musical rhythmic priming influences visual word processing during  silent reading, building on prior research showing similar effects for  spoken word processing.

The  experiment was well-designed and has been very clearly presented. I  have no major concerns, but list a few minor points below:

1.     2.3,  1st paragraph. Were these patterns previously tested for their strength  of metricality? It might be worth adding a few sentences justifying  their design. In particular, why didn't the structure of the two  sequences parallel each other? That is, why was the short-long sequence  not 100 ms tone - 100 ms rest - 200 ms tone - 200 ms rest?

Response:  The structure of the auditory sequences used in the present study was  based preliminary on behavioral data. Previous research showed that  native speakers of English have a cultural bias toward grouping a  sequence of tones differing in duration, into short-long patterns (e.g.,  Hay & Diehl, 2007; Iversen, Patel, Ohgushi, 2008).  They thus  conducted a series of behavioral pilot experiments with different  iterations of the tone sequences to determine which parameters would  provide consistent ratings of either long-short or short-long patterns.  This information has been added to the method section on page 4, lines  218-222.

2.     2.3,  4th paragraph. Why not include words that are also pronounced with  iambic stress in Southern American English and/or run a pronunciation  check with the participants? It's not clear why iambic words were  included in the set of visual targets at all, if they were not going to  be analysed, and showing the same effect for trochaic and iambic words  would have strengthened the conclusions.

Response:  The iambic words were included to introduce variability in the word  stimuli for the lexical decision task. Since participants were not made  explicitly aware of the rhythmic manipulation, we were concerned that  using only trochaic words would be too obvious at some point during the  experiment. Given the nature of the task, we primarily controlled for  lexical frequency of the iambic words and trochaic words to ensure that  participant would be familiar with them. We did not try to limit the  pool of iambic words to those that show a consistent pronunciation in  the south because those words tend to be of lower lexical frequency. We  agree with the reviewer that the use of a pronunciation check may be one  possible venue to examine iambic words in future work. We believe,  though, that the design and task may have to be modified to avoid  potentially making the participants aware of the underlying rhythmic  manipulation given that a pronunciation check, in combination with the  musical aptitude test, may point them toward the actual nature of the  study.

3.     4,  2nd paragraph. Even if participants were instructed not to read the  words out loud, they presumably used verbal imagery to say them  covertly/audiate. The way the words were presented (one word at a time,  with the stressed syllable underlined) likely encouraged this way of  reading. So, despite the visual presentation, the processing of the  words was likely still auditory. It seems to me that the experiment  shows how musical rhythm priming effects generalize to verbal imagery  rather how they generalize to the visual modality.

Response:  We are thankful to the Reviewer for these insightful remarks. In Figure  1 provided in the manuscript, the stressed syllable was underlined for  illustration purpose only, to help readers whose first language many not  be English. However, during the experiment, stressed syllables were not  indicated to the participants. All words and pseudowords were presented  in black lowercase characters on a white background. This information  has been clarified in the description of the experimental procedure on  page 5, lines 259-261.

Still,  we totally agree with the reviewer that this is unlikely a purely  visual effect. In fact, we propose in the discussion that the present  findings further support the idea that prosodic information is  internally and automatically generated during silent reading (See Breen  (2014) for a review). Implicit prosody closely relates to the notion of  verbal imagery and inner voice. We have expanded the discussion on  implicit prosody on page 9, line 543-553.   

4.     Lines 231-232, "...observed for mismatching target *words* was generated by musical prime *sequences*..."

Response:  We thank the reviewer for catching these two grammatical errors, which  have been corrected in the revised version of the manuscript (page 7,  line 340-341).

Round  2

Reviewer 1 Report

Thanks for revising the paper - I have found it much improved from the first version.

However, the authors have not fully addressed the issue raised in the previous review regarding tthe statistical analysis, which seems as if it could be straightforwardly obtained from the existing data.

In their response, the authors stated :

"the outcome measures are different for hypotheses 2 and 3 (speech rhythm sensitivity and reading achievement, respectively"

But this should be reversed, somewhat: hypotheses 2 and 3 have the same outcome (ERP negativity), and different predictor variables (AMMA and ACT).

From the paper, hypothesis 2: "we expected the amplitude of the negativity elicited by the cross-modal priming effect to correlate with individual scores on a musical aptitude test"

hypothesis 3: "we expected the amplitude of the negativity elicited by the cross-modal priming effect correlate with individual scores on the ACT reading test"

Although these are correlations, and could be described in either direction, clearly the authors are examining whether individuals with higher musical aptitude and/or reading ability display greater sensitivity to word stress.

This is precisely the case when a statistical analysis including *both* variables is needed, because the contribution of one variable (e.g., ACT) to the ERP may change when controlling for the other variable.

it is somewhat interesting to say that one outcome variable (ERP) is correlated (separately) with two different predictor variables, but it is *more interesting* to say whether those predictor variables independently contribute to the outcome variables.

Author Response

November 25, 2018

Dear Editors,

Please find attached our revisions for the manuscript entitled “Cross-Modal Priming Effect of Rhythm on Visual Word Recognition and its Relationships to Music Aptitude and Reading Achievement”. We are pleased thatReviewer 1 found that the manuscript was much improved and appreciate the additional clarifications regarding the analysis of the data. We are very thankful for the suggestions as we believe that the additional statistical analysis has allowed us to strengthen the discussion.

Please find below our detailed responses to the reviewer’s comments. We hope to have addressed all of your concerns and that you will find the revised manuscript suitable for publication.

Sincerely yours,
Tess Fotidzis, Heechun Moon, Jessica Steele, Cyrille Magne

_____________________________________________________________

Thanks for revising the paper - I have found it much improved from the first version. However, the authors have not fully addressed the issue raised in the previous review regarding tthe statistical analysis, which seems as if it could be straightforwardly obtained from the existing data.

In their response, the authors stated :
"the outcome measures are different for hypotheses 2 and 3 (speech rhythm sensitivity and reading achievement, respectively"
But this should be reversed, somewhat: hypotheses 2 and 3 have the same outcome (ERP negativity), and different predictor variables (AMMA and ACT).
From the paper, hypothesis 2: "we expected the amplitude of the negativity elicited by the cross-modal priming effect to correlate with individual scores on a musical aptitude test"
hypothesis 3: "we expected the amplitude of the negativity elicited by the cross-modal priming effect correlate with individual scores on the ACT reading test"
Although these are correlations, and could be described in either direction, clearly the authors are examining whether individuals with higher musical aptitude and/or reading ability display greater sensitivity to word stress. This is precisely the case when a statistical analysis including *both* variables is needed, because the contribution of one variable (e.g., ACT) to the ERP may change when controlling for the other variable.
it is somewhat interesting to say that one outcome variable (ERP) is correlated (separately) with two different predictor variables, but it is *more interesting* to say whether those predictor variables independently contribute to the outcome variables.

Response: Following the reviewer’s suggestion, we have performed a multiple regression analysis with ACTReading scores and AMMA Rhythm scores as predictors and the ERP negativity as the outcome measure. Accordingly, we have made the following changes to the manuscript:

A sentence reflecting this new analysis has been added to the Data Analysis section of the Methods (page 6, lines 295-297).

The result of the multiple regression analysis is reported on page 7 (lines 320-327 and Table 1). Please note that we have also streamlined the description of the correlation analyses to eliminate redundant wording (pages 6-7, lines 309-319).

The discussion has been revised on page 8 (lines 412-413) and page 9 (lines 628-651).1 
